# A Review of In Vitro Methods for Measuring the Glycemic Index of Single Foods: Understanding the Interaction of Mass Transfer and Reaction Engineering by Dimensional Analysis

Yongmei Sun [ID], Chao Zhong [ID], Zelin Zhou, Zexin Lei [ID] and Timothy A. G. Langrish *[ID]

Drying and Process Technology Group, School of Chemical and Biomolecular Engineering, Building J01, The University of Sydney, Camperdown, NSW 2006, Australia; ysun2550@sydney.edu.au (Y.S.); czho7722@uni.sydney.edu.au (C.Z.); zzho3224@uni.sydney.edu.au (Z.Z.); zlei7127@uni.sydney.edu.au (Z.L.)
* Correspondence: timothy.langrish@sydney.edu.au

**Abstract:** The Glycemic Index (GI) has been described by an official method ISO (International Organization for Standardization) 26642:2010 for labeling purposes. The development of in vitro methods for GI measurement has faced significant challenges. Mass transfer and reaction engineering theory may assist in providing a quantitative understanding of in vitro starch digestion and glycemic response from an engineering point of view. We suggest that in vitro GI measurements should consider the mouth and the stomach in terms of fluid mechanics, mass transfer, length scale changes, and food-solvent reactions, and might consider a significant role for the intestine as an absorption system for the glucose that is generated before the intestine. Applying mass transfer and reaction engineering theory may be useful to understand quantitative studies of in vitro GI measurements. The relative importance of reactions and mass-transfer has been estimated from literature measurements through estimating the Damköhler numbers ($Da$), and the values estimated of this dimensionless group (0.04–2.9) suggest that both mass transfer and chemical reaction are important aspects to consider.

**Keywords:** in vitro digestion; mass transfer; chemical reaction; glycemic index; starch hydrolysis; oral digestion; plant cell wall

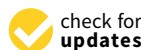



## 1. Introduction

The concept of Glycemic Index (GI) was suggested by Jenkins to classify carbohydrate-containing foods. GI is "an expression of the percentage of the area under the blood glucose response curve when taking the same amount of carbohydrate as glucose" [1]. It is a physiological way to explain how dietary carbohydrate impacts blood glucose. The GI value has a range between 1 and 100. Glucose, as the reference material, has a GI value of 100. A food with a lower GI value (≤54) raises blood glucose more slowly [2].

Low-GI diets help with the management of obesity, diabetes, and cardiovascular disease [3–6]. Low-GI foods must be identified by the method of ISO 26642:2010 for labeling purposes [7–9]. The ISO 26642:2010 test is an in vivo method that involves many voluntary participants, and the in vivo method is time consuming [10]. Ethical clearance is also necessary and may be another barrier for rapid trial and error testing for some foods during their development stages.

Researchers have investigated some in vitro methods for GI measurements of single foods. Single foods refer to non-processed foods, such as banana and carrot, which are the focus of in vivo methods for GI measurement. The in vitro digestion results for carbohydrates have been associated with GI values [11,12]. However, the correlations between in vivo and in vitro measurements are not consistently high [13,14], and other physiological factors, including the glucose tolerance of different individuals, and meal factors, such as the physical forms of the foods, can confound the relationship [15–17]. The current

in vitro method is only employed for food product development, but not for food labeling purposes [18,19].

This review aimed to understand the glycemic response (glucose measurement) from starch, which is the major glycemic carbohydrate in human foods [20,21]. The objectives were to (1) apply mass transfer theory to carbohydrate digestion to understand in vitro GI measurement, and (2) to use dimensionless groups, such as the Damköhler number (*Da*), and dimensional analysis to provide an engineering point of view about the importance of mass transfer and chemical reaction for the glycemic response in food digestion studies.

## 2. In Vivo Methods for GI Measurement

### 2.1. ISO 26642:2010 Method

An official method for measuring GI was issued by the International Organization for Standardization (ISO) in 2010 [22] after development and update [21,23]. The ISO method has been calibrated by three independent laboratories, Sydney University, Australia, and GI labs in Toronto, Canada, and Biofortis Merieux NutriSciences in Saint-Herblain, France. Using the 2010 ISO method is precise enough to differentiate a low-GI food from a high-GI food with a high probability (97–99%) [24].

The in vivo method of GI measurement is summarized in Table 1. Participant recruitment [24], test sample preparation [22], blood sample collection [22], and data analysis [25,26] are key steps for this ISO method.

**Table 1.** Description of the in vivo method for GI (Glycemic Index) measurement using the ISO (International Organization for Satandardization) 2010 method.

| Key Steps | Test Design | Reference |
|---|---|---|
| Participants | More than 10 people; No known food allergy; 18–35 years old; Non-smokers; Healthy (8 data/range of criteria). | [24] |
| Test samples | Reference food: 50 g glucose; Test food: 50 g carbohydrate containing; 250 mL water served. | [22] |
| Blood samples | Take blood samples at −10, −5, 15, 30, 45, 60, 90, 120 min. | [22] |
| Data analysis | Spectrophotometry or electrochemical detection-coupled enzyme systems. | [25] |

Prior to in vivo tests for GI measurement, test foods are usually analyzed to meet stringent nutritional criteria for energy (kJ or kcal), carbohydrate, saturated fats, sodium, and (in certain foods) fibre and calcium [7–9].

### 2.2. The Opportunities for Using Alternative Methods

The in vivo method of GI measurement has high accuracy and precision [23,24]. At the same time, it requires extra effort to manage the participants and obtain ethics clearance [22]. The cost of the in vivo test is relatively high [10], especially for food formulae, which are still under development. Other researchers have explored alternative methods to determine the GI values of foods, mainly in vitro methods. The GI value determined by an in vitro method is sometimes incorrect when classifying a low-GI food, and to label a high-GI food as a low-GI category is potentially harmful for people with diabetes [18]. So far, the GI measurement of single foods is still determined by in vivo methods [18].

Besides the glycemic effect, the in vitro method also covers the understanding of food nutrition and formulas, food digestibility and other health benefits by mimicking food digestion in living bodies. Many studies have focused on the chemical analysis of the food digestion process [20,27–29]. The chemical analysis, either for physical models or for mathematical models [30–34], has not always considered shear stresses and shear rates when

mimicking the digestion process [35]. A comprehensive review of all digestion models is covered by a related paper [35] (Sections 2 and 4 in [35]). Recent digestion models focus on mimicking the kinetics of food movement in the digestive system, as well as the physical processes during the peristaltic movement of digestion system. It is worthwhile to understand food digestion (especially carbohydrate digestion) from engineering perspectives to pave the way for an improved in vitro methods of GI measurement.

## 3. Food Digestion and Process

### 3.1. Carbohydrates

Starch is the majority carbohydrate in plants and is deposited in granules in most green plants. Its hydrolysis provides 40–80% of the total human energy intake [21,36]. It is found in many types of plant tissues and organs, such as seeds (e.g., cereal grains), roots (e.g., sweet potato), tubers (e.g., potato), stems (e.g., sago), leaves (e.g., tobacco), fruits (e.g., banana), and even pollen [37,38]. Starch is the dominant component of cereal grains, pluses, and tuber and root crops [37]. For instance, milled rice kernels contain up to 90% starch on a dry basis [39], maize kernels contain up to 80% starch [40], and potatoes contain 60–80% starch [41]. Besides starch-rich crops, pulse grains, such as legumes, have up to 53% starch [42]. Starch is a biopolymer. It contains two major components: amylose and amylopectin. Amylose is a mainly linear polysaccharide, which contributes up to 15–35% of the granules. Amylopectin, however, is a highly branched polysaccharide [37,38,43]. Amylose, containing α-1,4-linked d-glucopyranose and a few branches of α-1,6 linkages, has different properties to amylopectin with α-1,4-linked linear chains of different lengths, connected by about 5% α-1,6 branch linkages [38,44–46]. Amylose tends to produce tough gels and strong films, while amylopectin produces soft gels and weak films. It has been reported that a high amylose content in starch may help reduce the glycemic response and increase the blood glucose level slowly [47].

Recent studies have advanced the understanding of the starch features and the fine structure of amylose and amylopectin [37,48]. The importance of plant cell walls and the cellulose tissue structure have been noticed by several works [49–51], and these features have also been discussed in the studies of carbohydrate digestion [52–54]. The cell wall may be an important mass-transfer resistance from a mass transfer point of view. Starch granules transfer through the broken cell wall before its hydrolysis with digestive enzymes during the digestion processes. This section may be divided by subheadings. It should provide a concise and precise description of the experimental results, their interpretation, as well as the experimental conclusions that can be drawn.

### 3.2. Digestion Process

Studying human digestion was initiated in the medical field for diagnostic purposes [55]. Nowadays, it is an essential factor in the development of novel food products, as well as the testing of new pharmaceutical products [56,57].

The process of food digestion generally includes oral digestion, esophageal transit, gastric digestion, small intestinal digestion, and large intestinal fermentation. Oral digestion is the initial process to produce a bolus, which is a mass of chewed food [58,59]. The bolus has small particle sizes for safe swallowing [60,61]. Once a bolus is swallowed, it moves by esophageal peristalsis, as well as by the simple force of gravity when not lying down [62]. Then, the bolus passes into the stomach. The stomach plays the role as a food container, mixer, grinder, and sieve [63]. The majority of food breakdown happens in the stomach, where the bolus blends with gastric acid as well as digestive enzymes. The speed of food breakdown in the stomach is important to determine other digestion processes, such as gastric emptying, as well as nutrient absorption [57]. Food moves into the small intestine after it moves out of the stomach. Further food breakdown happens in the small intestine. The partially digested food from the stomach is broken down into small molecules to be absorbed and carried into the bloodstream. The large intestine is colonized

by microorganisms, which ferment food particles that have not been digested completely. Only water and fermentation by-products are absorbed in the large intestine [62].

Knowledge about the human digestion process may benefit from research in the medical and nutrition fields. However, the fundamental mechanisms are still not completely understood. For instance, Glycemic Index measurement is common in nutritional studies and is conducted by a physiological method [24,64]. The factors related to the glycemic response still require quantitative understanding in terms of the physical and chemical properties of foods. Mass transfer theory may help provide the quantitative analysis by understanding the processes of mass transfer and chemical reaction during food digestion processes. Such engineering perspectives may contribute to the fundamental understanding of food digestion.

### 3.3. Starch Digestion

The digestion process of starch can be partially quantified by the rate of starch loss, the rate of glucose appearance, and the rate of appearance of various oligosaccharides [65]. It has been stated that "Understanding the factors influencing starch digestion is best done through a causal, mechanistically based approach through the following paradigm: biosynthesis → growth and processing conditions → structure of starch and of starch-containing substances → digestion properties" [65].

In the human digestive system, starch is catalyzed by salivary amylase and pancreatic amylase, which are both $\alpha$-amylases (Enzyme Commission number is 3.2.1.1) [66]. Salivary amylase is the first enzyme for starch hydrolysis in the mouth [57,67]. This process occurs over a relatively short time (within one minute). When the bolus of food moves into the stomach, the action of $\alpha$-amylase slows down and the acid hydrolysis of starch increases. The hydrolysis of starch in the stomach may also be affected by the residual activity of salivary amylase, and the acidity of the stomach is likely to partly reduce the activity of the salivary amylase [68]. From the stomach to the duodenum, the bolus encounters $\alpha$-amylase the pancreatic secretion, which contains sodium hydrogen carbonate and $\alpha$-amylase. Sodium hydrogen carbonate neutralizes the acidic fluid from the stomach to a pH of about 8 [69]. The continues the catalysis of starch into disaccharides and oligosaccharides. The oligosaccharides, such as $\alpha$-limit dextrins, small linear oligomers, and larger $\alpha$-glucans are not absorbed into the blood stream until their further hydrolysis to glucose. In the small intestine, enzymes, including mucosal maltase–glucoamylase and sucrase–isomaltase, catalyze the oligosaccharides into single glucose [65,68].

After a meal, the peak plasma glucose response usually occurs within the first hour, and the glucose level increase seldom lasts more than two hours. This observation puts a clear emphasis on the mouth, salivary fluids, and the stomach as features that may be very important in the glycemic response. At the same time, the pancreas secretes insulin and inhibits the release of glucagon, so that the glucose is normally taken up by muscle and fat tissue. Plasma glucose levels have a range between 3.3 and 8.3 mmol/L, providing body energy for the organs and tissues. However, high postprandial glucose levels are related to the development of Type 2 diabetes and/or cardiovascular disease in susceptible persons [3,70,71]. People with diabetes have a high blood glucose level (hyperglycemia) due to deficiencies in insulin secretion or in insulin action [15,71]. The current in vivo method to determine the glycemic response of a meal/food is to measure small numbers of blood samples from the finger over a period of two hours. The Glycemic Index is then calculated to classify carbohydrate-rich foods [24]. Understanding starch digestion and absorption of starch-derived glucose may help in the maintenance of stable plasma glucose levels.

### 4. In Vitro Digestion

#### 4.1. Digestion Models

Food digestion studies can be done in vivo and in vitro. In vivo studies tend to be more realistic, however, they are complex, expensive, and can have ethics-related issues [57].

In vitro digestion studies started a long time ago [72]. In general, models for in vitro digestion mimic in vivo conditions, such as the chemicals involved, the reaction environment, the materials, and the motility (the movement of food), including the simplified model of Edwards and co-workers described in [73]. For example, refs [11,73] have predicted glycemic response using static models. The Englyst method and the simplified model proposed by Edwards are both static models. There should be no intrinsic problems with static models for in vitro GI assessments, provided that due care is taken to operate them in such a way that the mass-transfer coefficients [74] are representative of those in in vivo systems. The conditions in each step of an in vitro model represent pH changes and digestion times throughout the digestion processes in the oral cavity, the stomach, the intestine, and the colon [75].

Oral digestion is often difficult to mimic [76]. Some oral processing models investigate fracture mechanics [77], which is a physical way to understand the breakage of solid foods under large deformations [78]. Other studies simplify the oral processing by mincing the test samples in a commercial blender [79]. Oral processing combines mastication, lubrication, conveyance of food particles in a bolus, and swallowing [77]. It is not only a physiological process dominated by the central nervous system, but also a physical process modulated by the mechanical and geometrical properties of the food [80].

Food rheology, transport phenomena, and particle size distribution are discussed in many studies of oral processing [81–83]. Researchers have noted that the important roles of mastication and salivary amylase in determining the postprandial glycemic response [84,85]. Several clinical studies have also emphasized the connection between the oral digestion process and the glycemic response [86–88]. Though the aims of these researchers have not been to develop an in vitro test method of GI measurement, their test results have provided relevant evidence for the involvement of the oral digestion process in overall digestion models [89].

Stomach models often reflect pH changes (from pH 5–7 in mouth to pH 1–3 in the stomach) and the dynamics of stomach movement, such as the MMC (Migrating Motility Complex), which is the pattern of physical movements caused by electrical activity [90]. The latest stomach models simulate the stomach's dynamics and have controlled and/or standardized conditions [56,91–95]. However, improving and understanding the mechanical aspects of models, such as internal flow patterns, shear stresses, and shear rates are still research gaps for improving in vitro digestion models [35].

Gastrointestinal models have sometimes focused on the benefits of eating probiotics [96,97]. Most recently, gastrointestinal models, from static noncompartmental to dynamic multicompartmental ones, have been broadly used as alternatives to in vivo assays in various fields such as pharmacology, toxicology, microbiology, and nutrition [30,98]. Artificial Intelligence (AI) has great potential to simulate human experience using computer programs. Artificial neural network models have been discussed in pharmaceutical research [99]. AI may provide many promising possibilities in this research such as data analysis and simulation tools in the future.

*4.2. In Vitro Starch Digestion*

Starch digestion studies with in vitro digestion models provide in vitro methods to understand the glycemic response of a food and potentially develop in vitro tests for Glycemic Index measurement. For starches and carbohydrates, the structure, amylose content, amount of moisture, and fiber content are important in determining the glycemic response [100–103]. Besides the properties of starch, there are several factors affecting the glycemic response or Glycemic Index determination during digestion: (1) mechanical aspects, such as the shear stress in chewing, the flow patterns during swallowing and stomach mixing, and the shear rate during peristalsis, which may change the rate of food breakdown and the development of the particle size distribution; (2) food bolus transportation and rheology in the gastrointestinal tract, which may affect transport parameters, such

as the mass-transfer coefficient; (3) the kinetics of starch hydrolysis and amylase (salivary amylase and pancreatic amylase) catalysis [104,105].

Table 2 summarizes in vitro studies of glycemic response through starch/carbohydrate digestion models. The focus is to review relevant and well-developed models, which mimic the in vivo method of Glycemic Index measurement [24] and to illustrate research gaps for further studies.

**Table 2.** Summary of glycemic response studies using in vitro starch digestion models.

| In Vitro Digestion Model for Starch Digestion | | | How It Works | References |
|---|---|---|---|---|
| Oral models: | AM2 | Physical model | Simulate human masticatory behavior and generate a food bolus with similar granulometric characteristics. | [106,107] |
| | B-SPH | Mathematical model | Simulate several complicated aspects of mastication. | [108] |
| Gastrointestinal models: | TIM-Carbo | Physical model | Test the digestibility of carbohydrates through the stomach and small intestine. Use the availability of monosaccharides to predict glycemic response. HOMA for Data process. | [109,110] |
| | DGM | Physical model | Monitor the changes in the ratio of glucose:starch and total starch effect of particle size on starch hydrolysis from Durum wheat. | [111,112] |
| | CSTR PFR | Physical model | Investigate the digestion and absorption of starch and glucose in the small intestine. Understand the relative effect of gastric emptying time and luminal viscosity on the rate of glucose absorption. | [113] |

An oral model, the AM2 (Artificial Masticatory Advanced Machine), was developed to simulate human masticatory behavior. It produces a food bolus with similar granulometric characteristics to the real food bolus in human digestion. Different variables may be altered in the model, such as the amplitude of the mechanical movements, mimicking the vertical and lateral movements of the lower jaw in humans, the masticatory cycles and forces, the temperature of the mastication chamber and the components of saliva [106]. The B-SPH (Biomechanical-Smoothed Particle Hydrodynamics) model was shown to simulate the complexity of mastication, which includes particle size changes and large strain behavior due to softening by heating, and solid and liquid food component interactions [108]. These parameters also provide test conditions for the Glycemic Index measurement of food.

Some gastrointestinal models, which were initially developed two to three decades ago, have become commercially available for in vitro digestion research or nutritional studies. For example, the TIM-Carbo (TNO Gastro-Intestinal Model), the DGM (Dynamic Gastric Model) and the CSTR (Continuous Stirred Tank Reactor) with the PFR (Plug Flow Reactor) have been broadly used in the development of food and pharmaceutical products, mainly as screening tools [109–113]. A very realistic stomach model, the DIVHS (Dynamic In Vitro Human Stomach) has recently been employed to understand yogurt digestion [114]. When applying these models for glycemic response studies, the food oral process has often been replaced by a blender or a mincer instead of mastication simulators [93,115].

Although there are some very sophisticated in vitro digestion models, many studies of starch digestion and/or glycemic response have been conducted in beaker and stirrer systems [20,116,117]. An advantage of using beaker and stirrer systems is the flexibility to change different factors, which affect the glycemic response of starch digestion. Several reviews of digestion models have been reported recently in the literature [34,35,118]. Current models may be a good start for developing our fundamental understanding of Glycemic Index measurement. The research focus in this work has been the reaction kinetics of starch hydrolysis and the associated mass-transfer processes. Both beaker–stirrer systems and oral digestion models may be used in future studies.

*4.3. Engineering Perspectives*

The starch hydrolysis begins in the mouth, as shown in Figure 1. During oral processing, about 50% of bread and 25% of pasta starch is hydrolyzed and is transformed into smaller molecules (particle size reduction) in a short period of time [86,119]. The different rates of starch hydrolysis are caused by structural differences in the solid foods [67,86]. In an in vitro study, it was found that, in less than 10 s of blending with saliva, custard showed an approximately tenfold decrease in its viscosity [82,120,121]. The $\alpha$-amylase enzyme is most active at a pH of 7.4. Thus, the enzyme is fully functional inside the mouth. However, it is inactivated in the stomach due to the presence of gastric acid [122].

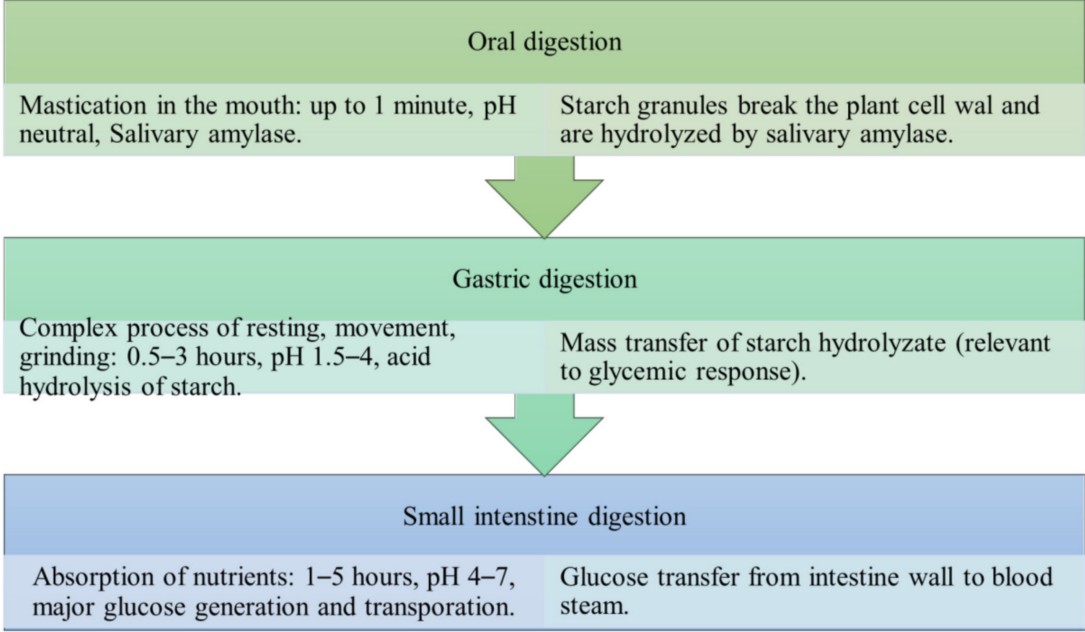

**Figure 1.** Diagram of engineering food digestion focusing on starch hydrolysis.

The interaction between $\alpha$-amylase and starch starts almost immediately after food ingestion. Free glucose is generated, together with other sugars, such as maltose and oligosaccharides [57,67,123]. Considering the digestion procedure for food carbohydrates, and the typical sampling intervals (15, 30, 45, 60, 90, and 120 min after ingestion) for blood glucose measurements (in the in vivo method) [24,124], the glucose from starch hydrolysis, starting in the mouth, may play a crucial role in the glycemic response and increase the amount of blood glucose within the first hour of digestion. Most of the starch breaks down in the stomach and is digested by pancreatic amylase. For the measurement of the Glycemic Index, starch hydrolysis by $\alpha$-amylase in both the mouth and the stomach is important because the test monitors the blood glucose levels during the first two hours after food consumption, over short time intervals (15 or 30 min) [24].

Current studies on glycemic response and/or starch hydrolysis have focused more on gastrointestinal digestion and nutrition absorption. However, the glucose generated in the small intestine by glucosidases (sucrase–isomaltase and maltase–glucoamylase) is mainly available as an energy source taken up by muscle and fat tissue [15,71,123]. It may be worthwhile to extend the investigation into oral digestion processes and consider oral digestion along with gastrointestinal digestion for starch hydrolysis, particle size reduction, rheological changes, and mass transfer. The topics of starch hydrolysis, rheological changes, and particle size reduction during food digestion are intensively covered by recent research. However, applying mass transfer for understand food digestion and starch digestion is a relatively novel perspective for glycemic response and in vitro GI measurement.

Mass transfer theory has been linked to food digestion [125], and mass-transfer coefficients are seldom reported as individual values. There is limited research on, or examples

of, studying mass-transfer coefficients in food digestion studies [126], despite the relevance of mass-transfer coefficients to the estimation of transfer rates for the solutes that are food components. Mass transfer coefficients are usually correlated with the use of dimensionless numbers and dimensional analysis to reduce the dimensionality in correlations or empirical equations that describe the results of experimental studies [127]. Table 3 summarizes dimensionless groups that may be relevant in food digestion studies. These dimensionless numbers are commonly correlated in mass transfer using equations of the following basic form:

$$Sh = a\, Re^b Sc^c C \tag{1}$$

**Table 3.** Dimensionless groups involved in food digestion.

| Dimensionless Group | Equation | Physical Meaning | Description | Ref. |
|---|---|---|---|---|
| Sherwood | $Sh = \frac{kl}{D}$<br>$k$ = mass transfer coefficient (m/s);<br>$l$ = length (m);<br>$D$ = mass diffusivity (m$^2$/s). | $\frac{\text{convective mass transfer rate}}{\text{diffusion rate}}$ | Mass transfer | [128] |
| Reynolds | $Re = \frac{\rho u l}{\mu}$<br>$\rho$ = density (kg/m$^3$);<br>$u$ = velocity (m/s);<br>$l$ = length (m);<br>$\mu$ = viscosity (Pa.s). | $\frac{\text{inertial forces}}{\text{viscous forces}}$ | Forced convection, fluid mechanics, heat transfer and mass transfer | [129] |
| Schmidt | $Sc = \frac{\mu}{\rho D}$<br>Symbols defined above. | $\frac{\text{momentum diffusivity}}{\text{mass diffusivity}}$ | Fluid mechanics and mass transfer | [130] |
| Damköhler | $Da = k_r C_0^{n-1} \tau$<br>$k_r$ = reaction rate constant (mol$^{1-n}$ m$^{3(1-n)}$/s);<br>$C_o$ = initial concentration (mol/m$^3$);<br>$n$ = reaction order;<br>$\tau$ = mean residence time (s). | $\frac{\text{reaction rate}}{\text{convective mass transport rate}}$ | Reaction engineering and mass transfer | [131] |

The Damköhler number (*Da*) is the ratio of the chemical reaction rate to the transport phenomena (convective mass transfer) rate occurring in a system [132]. It may also be calculated as the ratio of a transport time to a reaction time [133]. If *Da* >> 1, the reaction rate is much greater than the mass transfer rate. By contrast, if *Da* << 1, mass transfer occurs much faster than the chemical reaction [134]. The experimental approaches used in the past to obtain mass-transfer coefficients and reaction rates have been reviewed in [125], and the *Da* has been reported experimentally by [133,135]. For example (Figure 4.6 from [136]), in a small-scale biocatalytic reactor, where the enzyme is immobilized, at the time constant $\tau$ = 10 min and width of the reactor W = 200 μm, when *Da* = 100, the maximum reaction rate is mass transfer limited; when *Da* = 0.01, diffusivity is much greater. In the biocatalytic field, the *Da* has been used to predict mass-transfer limitations [135]. Another example is combustion chemistry, such as the oxidation of carbon and hydrogen from biomass fuel. A MILD (Moderate or Intense Low Oxygen Dilution) combustion model for low excess oxygen concentrations has used the *Da* and the Reynolds number to understand chemical reactions in the flames. In the reaction zone, the maximum *Da* (reaction rates) are higher and the Reynolds numbers (mass transfer rates) are lower than in the rest of the flow in combustors (Figure 2 from [137]). This concept of relatively high reaction rates in combustors when compared with the transport rates for reactants and products (high *Da*) is also a basis for the "mixed is burnt" combustion model that has been used in Computational Fluid Dynamic modeling of combustion devices [138–141].

For starch hydrolysis, where starch is catalyzed by amylase into maltose and other oligosaccharides by an irreversible reaction, the *Da* can be calculated from the following equation [131]:

$$Da = K_r C_0^{n-1} \tau \tag{2}$$

Here, *Kr* is the reaction rate constant ($s^{-1}$), $C_0$ is the initial concentration, *n* is the reaction order, and $\tau$ is the mean residence time (s), if the mean residence time is characteristic of the mass-transfer rate in the solution. This equation is used for a convective flow system. The exact formula for the *Da* may vary due to the various reaction rate law equations that apply to different reactions.

The kinetics of starch hydrolysis often follow a first order equation [142], as shown below:

$$C = C_\infty \left(1 - e^{-kt}\right) \tag{3}$$

where *C* is the concentration of product (such as glucose, maltose, or other oligosaccharides) at time *t*, $C_\infty$ is the equilibrium concentration, *k* is the kinetic constant or the reaction rate constant, and *t* is the chosen time. The *k* value has a range from $10^{-5}$ to $10^{-3}$ $min^{-1}$, depending on the concentration and type of the enzyme [143].

### 4.4. Estimating Damköhler Numbers from Literature Reaction Data

Here, we analyzed reaction data for starch hydrolysis from the literature, specifically data from [113,132,144] relative to the mass-transfer rates in their situations, in order to estimate the ranges of *Da* in these situations.

Table 4 summarizes the estimated *Da* using the experiments on starch hydrolysis by $\alpha$-amylase from [132]. This study has investigated the starch hydrolyzates and reaction rate during wheat starch hydrolysis with $\alpha$-amylase from *Bacillus licheniformis*, considering effects such as the starch pre-treatment, the enzyme addition point, and the hydrolysis conditions. An example of the *Da* calculation is given below, using the test system with an amylase/starch ratio (% *w/w*) of 0.1 in 5% (*w/w*) wheat starch solution, as shown for high-temperature gelatinized starch in "Figure 1. dextrose equivalent as a function of the hydrolysis time for three different enzyme/substrate ratios after HT gelatinization. Hydrolysis conditions: $\alpha$-amylase from *B. licheniformis*, 50 °C, 5 *w/w* % wheat starch in water" of [132].

**Table 4.** *Da* involved in starch hydrolysis by $\alpha$-amylase (data from [132]). The residence times were all 60 min.

| 5% (*w/w*) Wheat Starch Solution | Reaction Rate Constant ($min^{-1}$) | *Da* |
|---|---|---|
| Amylase/Starch ratio (% *w/w*): 0.1 | 0.003 | 0.19 |
| Amylase/Starch ratio (% *w/w*): 1.0 | 0.011 | 0.65 |
| Amylase/Starch ratio (% *w/w*): 10 | 0.007 | 0.45 |

In the test results of Figure 1 from [132], a hydrolysis time 60 min has been chosen, where *C* = 14 (units of D.E, dextrose equivalent) at *t* = 60 min and $C_\infty$ = 38 D.E. Using Equation (3),

$$
\begin{aligned}
14 &= 38\left(1 - e^{-k60}\right) \\
e^{-k60} &= 1 - 14/38 \\
&= 0.631579 \\
-k60 &= \log 0.631579 \\
&= -0.19957 \\
\text{So, } K_r C_0^{n-1} &= -0.19957 / -60 \\
&= 3.326 \times 10^{-3} \ min^{-1}.
\end{aligned}
$$

According to Equation (2), with a mean residence time of 60 min.

$$\begin{aligned} Da \ &= K_r C_0^{n-1} \tau \\ &= 3.326 \times 10^{-1} \text{min}^{-1} \times 60\text{min} \\ &= 0.19. \end{aligned}$$

A key assumption, in this equation, is that the mean residence time is characteristic of the mass-transfer rate, which is possibly debatable. Alternatively, the *Da*, being the ratio of the reaction rate to the mass-transfer rate, may be used in the following way. For the previous calculation, the reaction rate ($3.326 \times 10^{-3}$ min$^{-1}$= $5.54 \times 10^{-5}$ s$^{-1}$) may be compared with a characteristic mass-transfer rate, assuming a length scale of z m and a convective mass-transfer coefficient of $2 \times 10^{-5}$ m s$^{-1}$ [74,145]. The range of mass-transfer coefficients has been addressed in a recent paper [74], which includes the value used here, and the length scale for the calculation of mass-transfer coefficients is discussed in Section 8.3.1 (dimensionless numbers) of [125]. The time constant for mass transfer would then be z m/($2 \times 10^{-5}$ m s$^{-1}$), so that the *Da* is given by the equation:

$$\begin{aligned} Da &= 5.54 \times 10^{-5} \text{ s}^{-1} \ [\text{z m}/(2 \times 10^{-5} \text{ m s}^{-1})] \\ &= 2.77 \text{ z} \end{aligned}$$

In this case, a range of *Da* numbers from 0.1–1 would correspond to length scales for mass transfer of 0.056 to 0.56 m. A length scale of 0.056 m is certainly credible in this situation, since the diameter of the anchor stirrer used by [132] was 52 mm or 0.052 m. In other words, if we use a length scale for mass transfer (z) equal to the stirrer diameter of 0.052 m (52 mm), we obtain a *Da* number of 0.14. This is a similar value to the value (0.19) obtained from the previous perspective on the calculation.

In a system consisting of 5% (*w/w*) wheat starch solution, a higher amylase concentration (amylase/starch ratio (% *w/w*): 1.0) results in a higher reaction rate (0.011 vs. 0.003), thus giving a larger *Da* number (0.65 vs. 0.19). However, increasing the amylase concentration from amylase/starch ratio (% *w/w*) of unit (1.0) to amylase/starch ratio (% *w/w*) of 10 does not result in a higher reaction rate and a higher *Da* number.

Table 5 summarizes the estimated *Da* using the experiments on starch digestibility by $\alpha$-amylase from [144]. This study has emphasized that the degree of starch gelatinization is a major factor in the digestion rate. For the estimation of the *Da*, the reaction rate has been fitted in the literature using Equation (3). In the test results of "Figure 1. digestogram of potato starch samples with different degree of gelatinization" from [144], the *Da* number has been calculated according to Equation (2), for a residence time of *t* = 60 min.

**Table 5.** *Da* involved in starch digestibility by $\alpha$-amylase (data from [144]). The residence times were all 60 min.

| Potato Starch Solution (about 20% *w/w*) | Reaction Rate Constant (min$^{-1}$) | *Da* |
|---|---|---|
| Degree of gelatinization (%): 100 | 0.0496 | 2.9 |
| Degree of gelatinization (%): 34 | 0.0198 | 1.2 |
| Degree of gelatinization (%): 0 | 0.0118 | 0.7 |

In this situation, a higher degree of gelatinization results in a higher reaction rate, thus a larger *Da* number. A high degree of gelatinization may break the cell walls of starch granules and increase the mass transfer rate, while at the same time, the reaction rate also increases dramatically [49,52].

Another example in Table 6 summarizes the estimated *Da* from the intestine model for starch hydrolysis by $\alpha$-amylase from [113]. In this model simulation, starch is hydrolyzed by $\alpha$-amylase into glucose, then the glucose is absorbed into the small intestine. The *Da* has been calculated as the ratio of the reaction rate to the mass-transfer rate. According to "Figure 6. the effects of the gastric emptying rate, the mass transfer rate, and the reaction

rate for starch hydrolysis on the absorption of glucose" from [113], the mass-transfer rate has been found to be $0.5 \times 10^2$ min$^{-1}$, so the reaction rate has been obtained for the estimation of $Da$.

**Table 6.** $Da$ involved in starch hydrolysis by an intestinal model (data from [113]). The mass transfer rates were all $0.5 \times 10^2$ min$^{-1}$.

| Fraction of Glucose Absorbed in the Intestine Model Simulation | Reaction Rate (min$^{-1}$) | $Da$ |
|:---:|:---:|:---:|
| 0.2 | 2 | 0.04 |
| 0.6 | 7 | 0.14 |
| 0.8 | 14 | 0.28 |
| 1.0 | 25 | 0.50 |

This model assumes that the starch is intact when it arrives at the small intestine, which does not consider the starch hydrolysis by salivary amylase. For the glycemic response, however, starch hydrolysis in the mouth is very important in terms of the digestion time as well as the glucose generated. Greater starch hydrolysis gives more glucose absorption. When the fraction of glucose absorbed increases from 0.2 to 1.0 in the model, the $Da$ number rises almost tenfold.

The interaction between gastric emptying and glycaemia discussed by [146] provides some further support for our suggestion that the glycemic response may start in the mouth, where a significant part of the starch hydrolysis may occur in the stomach. One reference [146] indicates that gastric emptying has a significant influence on the maximum blood glucose levels after eating, being responsible for about 35% of the variance in these levels. They also point out that the rate of emptying is, in turn, affected by acute changes in glycaemia. Hence, it is clear that the residence time of food in the stomach affects the glycemic response significantly, and that the human body has a feedback loop where the body attempts to modulate the glycemic response by changing the residence time in the stomach.

In these three studies, the general range of $Da$ numbers have ranged from 0.04 (lowest value, [113]) to 2.9 (highest value, [144]). This situation means that the reaction rates are between 4 and 290% of the mass transfer rates, so both the reaction kinetics and the mass transfer are important in this situation. In general, the $Da$ number indicates a balance of chemical reaction and mass-transfer phenomena. A $Da$ of unity (1) means that the reaction rate equals the mass-transfer rate. It is therefore possible for a reaction to have a $Da$ range that spans unity, because it is possible to change the mass-transfer rate by changing the operating conditions, relative velocity between the food and the digestive juices, and the Reynolds numbers ($Re$) for the digestion [125], while the reaction rate may be changed by altering the ratios of reactants and the reaction temperature. Hence the ratio of the reaction rate to the mass-transfer rate, which is the $Da$, may be changed by altering the parameters just mentioned to change each of these rates (reaction and mass transfer) over a range that may include unity. This dimensionless number provides a quantitative tool to understand the process of both chemical reaction and mass-transfer, such as starch digestion.

Mass transfer theory is also applicable for the study of in vitro drug digestion, which focuses on changing the resistance to mass transfer to improve bioavailability and lower dosages of drugs while retaining their effectiveness. Dimensional analysis and dimensionless numbers may be useful to provide a quantitative understanding of in vitro drug digestion such as insulin oral administration [147].

## 5. Conclusions

In vivo GI measurement and human digestion studies provide critical observations, where we have reviewed and assembled many pieces of relevant evidence: (1) the Glycemic Index is typically measured in vivo over a two-hour period following food ingestion [24]; (2) the function of the mouth appears to be to cut up the food and inject salivary amylase [80]

before transporting the food to the stomach; (3) the food remains in the stomach for periods ranging from 30 min (liquids) to 2–3 h (solids) [35,124]. These pieces of evidence are consistent and are backed by peer-reviewed literature. Hence, it would appear to be reasonable to suggest that the Glycemic Index is significantly generated through mouth action and reactions in the stomach, with the function of the small intestine being to absorb any glucose units that are generated in the mouth and stomach. This consideration, in turn, suggests that studies of in vitro GI measurements should consider the mouth and the stomach for the fluid mechanics, mass transfer, particle size reduction and any reactions, and might consider the intestine as mainly an absorption system for the glucose that is generated before the intestine. Therefore, studies of in vitro GI response that consider mainly the small intestine might be usefully extended to include the mouth and the stomach [148].

Mass transfer theory, which considers the kinetics of mass component movement (solutes) in the solution (solvents), has been linked to food digestion [125], but only limited research has quantified mass-transfer coefficients [126]. When considering in vitro starch digestion and the glycemic response, mass-transfer resistance needs to be considered during quantitative studies because a starch granule must overcome the cell wall (cellulose) barrier before hydrolysis with amylase occurs [49]. The human body does not generate enzymes to digest cellulose. The cellulose (in the cell wall) in foods must be broken down by chewing, peristaltic movement, and stomach juice, so that the food nutrients can be released [149]. An important contribution of this work is that the *Da*, which are the ratios of the reaction rates to the mass-transfer rates, have been calculated for reaction rate data on starch hydrolysis from three literature papers [113,132]. The range of *Da* values estimated from the reaction rate data in these works has ranged from 0.04 to 2.9, showing that both mass transfer and chemical reaction are likely to be important in starch hydrolysis in typical situations.

**Author Contributions:** Conceptualization, Y.S. and T.A.G.L.; methodology, T.A.G.L. and Y.S.; software, Y.S. and T.A.G.L.; validation, Y.S., T.A.G.L., C.Z., Z.Z. and Z.L.; formal analysis, Y.S. and T.A.G.L.; investigation, Y.S. and T.A.G.L.; resources, T.A.G.L.; data curation, T.A.G.L., Y.S., C.Z., Z.Z. and Z.L.; writing—original draft preparation, T.A.G.L., Y.S., C.Z., Z.Z. and Z.L.; writing—review and editing, Y.S. and T.A.G.L.; visualization, T.A.G.L., Y.S., C.Z., Z.Z. and Z.L.; supervision, T.A.G.L.; project administration, Y.S. and T.A.G.L.; funding acquisition, T.A.G.L. All authors have read and agreed to the published version of the manuscript.

**Funding:** This research received no external funding.

**Acknowledgments:** The authors acknowledge support by an Australian Government Research Training Program (RTP) Scholarship and University Scholarships for Yongmei Sun, Zelin Zhou and Chao Zhong and a top-up scholarship for Chao Zhong from the Centre for Advanced Food Engineering (CAFÉ) in the Faculty of Engineering at The University of Sydney.

**Conflicts of Interest:** The authors declare that they have no known competing financial interests or personal relationships that could have appeared to influence the work reported in this paper.

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
