# Peer review of "A Review of In Vitro Methods for Measuring the Glycemic Index of Single Foods: Understanding the Interaction of Mass Transfer and Reaction Engineering by Dimensional Analysis"

_processes, doi:10.3390/pr10040759_

Round 1

Reviewer 1 Report

In the manuscript "A Review of In-Vitro Methods for Measuring the Glycemic Index of Single Foods" by Yongmei Sun the Mass transfer and reaction theory is used to deepen the quantitative understanding of in-vitro starch digestion and glycemic response from the engineering point of view.

The second version of the manuscript, without the typographical errors (probably due to the file's upload process), appears clear and of great interest, well written and definitely can deserve publication in Processes.

There are only minor concerns:

  • The title suggests that the manuscript is only a revision of the literature data and not a complex interpretation and the formulation of a new hypothesis. The novelty and the originality of the paper should be better evidenced;
  • The section "food digestion process" should be reduced, e.g. avoiding description of the well-known biochemistry of carbohydrates.
  • The sentences from lines 61 to 63 seem to be redundant and not clear;
  • The use of the abbreviation Da instead of Damköhler number should be standardized consistently along with all the text;
  • In taxonomy, the genus and the species are usually written in italics;
  • In table 3, more or less at the level of citation 131, a semicolon is misplaced.

Reviewer 2 Report

This manuscript reviews the state-of-the-art in in-vitro determination of GI for foods, with special emphasis on the relevance of correctly modelling the mass-transfer in starch dissolution dynamics. I am not an expert in physiology or in the modelling of starch metabolism, however, I find the review highly informative and interesting. In my view, this type of attempts to connect the nutritional aspects with engineering methodology is highly promising. Some more specific comments/suggestions/questions below:

  • To me, the paper reads as being somewhere in-between a traditional review and a research paper or thinking piece. In contrast to a classical review it adds new calculations (Da). The aims (lines 47-28) reads more like a scientific paper. Perhaps the aims could be reformulated to clarify that the aim is to present state-of-the-art and point out interesting directions for developments (or some other better wording)?
  • When first reading the paper, it was not clear to me that the calculation of Da on these starch dissolution data was new to this paper. It is clearly stated in the conclusion. Perhaps this could be clarified earlier?
  • From my perspective, it should not be at all controversial that fluid mechanics and mass transfer dynamics must be considered in understanding/modelling the GI response (but again, I am an engineer, not a nutritionist). Yet, I get the impression that the authors feel a need to argue very much for this in the text. Is this something that is controversial in the GI literature? Perhaps this is an issue that could be expanded on?

Minor and formatting comments

  • There are several faulty links in the references handling (in the version I received). For example, lines 57, 288, 328,371…
  • Online 356, there are two missing superscripts (10-5 and 10-3).
  • On line 356, the units for the reaction rate constant is 1/min. This only applies if reaction order n = 1? The correct expression should be the one used in Table 3.
